# Neocortical layer-5 tLTD relies on non-ionotropic presynaptic NMDA receptor signaling

Aurore Thomazeau[1,2†], Sabine Rannio[1,3†], Jennifer A Brock[1,3], Hovy Ho-Wai Wong[1,4], Per Jesper Sjöström[1*]

[1]Centre for Research in Neuroscience, Brain Repair and Integrative Neuroscience Program, Departments of Neurology & Neurosurgery and Medicine, The Research Institute of the McGill University Health Centre, Montreal General Hospital, Montreal, Canada; [2]Université Côte d'Azur, CNRS UMR7275, Institute of Molecular and Cellular Pharmacology, Valbonne, France; [3]Integrated Program in Neuroscience, McGill University, Montreal, Canada; [4]Gerald Choa Neuroscience Institute & School of Biomedical Sciences, The Chinese University of Hong Kong, Hong Kong, China

*For correspondence: jesper.sjostrom@mcgill.ca

†These authors contributed equally to this work

Competing interest: The authors declare that no competing interests exist.

## eLife Assessment

By using sparse Cre-dependent deletion of GluN1 subunit, in vitro quadruple patch clamp recordings, and pharmacological interventions, the authors show that spike timing dependent plasticity at between L5 synapses in the mouse visual cortex is: (i) dependent on presynaptic NMDA receptors; (ii) mediated by non-ionotropic NMDA receptor signaling, and (iii) reliant on presynaptic JNK2/Syntaxin-1a interactions. These **fundamental** findings advance our understanding of the molecular mechanisms underlying spike time dependent plasticity. The data are **compelling** and are supported by the elegant application of sophisticated experimental approaches.

**Abstract** In the textbook view, NMDA receptors (NMDARs) act as coincidence detectors in Hebbian plasticity by fluxing $Ca^{2+}$ when simultaneously depolarized and glutamate bound. Hebbian coincidence detection requires that NMDARs be located postsynaptically, but enigmatic presynaptic NMDARs (preNMDARs) also exist. It is known that preNMDARs regulate neurotransmitter release, but precisely how remains poorly understood. Emerging evidence suggests that NMDARs can also signal non-ionotropically, without the need for $Ca^{2+}$ flux. At synapses between developing visual cortex layer-5 (L5) pyramidal cells (PCs), preNMDARs rely on $Mg^{2+}$ and Rab3-interacting molecule 1αβ (RIM1αβ) to regulate evoked release during periods of high-frequency firing, but they signal non-ionotropically via c-Jun N-terminal kinase 2 (JNK2) to regulate spontaneous release regardless of frequency. At the same synapses, timing-dependent long-term depression (tLTD) depends on preNMDARs but not on frequency. We, therefore, tested in juvenile mouse visual cortex if tLTD relies on non-ionotropic preNMDAR signaling. We found that tLTD at L5 PC→PC synapses was abolished by pre- but not postsynaptic NMDAR deletion, cementing the view that tLTD requires preNMDARs. In agreement with non-ionotropic NMDAR signaling, tLTD prevailed after channel pore blockade with MK-801, unlike tLTP. Homozygous RIM1αβ deletion did not affect tLTD, but wash-in of the JNK2 blocker SP600125 abolished tLTD. Consistent with a presynaptic need for JNK2, a peptide blocking the interaction between JNK2 and Syntaxin-1a (STX1a) abolished tLTD if loaded pre- but not postsynaptically, regardless of frequency. Finally, low-frequency tLTD was not blocked by the channel pore blocker MK-801, nor by 7-CK, a non-competitive NMDAR antagonist at the co-agonist site. We conclude that neocortical L5 PC→PC tLTD relies on non-ionotropic preNMDAR signaling via JNK2/

STX1a. Our study brings closure to long-standing controversy surrounding preNMDARs and highlights how the textbook view of NMDARs as ionotropic coincidence detectors in plasticity needs to be reassessed.

## Introduction

Synapses continuously remodel in response to neuronal activity. Such synaptic plasticity is thought to underlie information storage (*Bliss and Collingridge, 1993*; *Malenka and Bear, 2004*; *Nabavi et al., 2014*) as well as developmental circuit refinement (*Cline, 1998*; *Katz and Shatz, 1996*; *Song and Abbott, 2001*), an idea often attributed to *Hebb, 1949*. More recent work tends to emphasize the role of temporal ordering of activity in determining plasticity, a notion called spike timing-dependent plasticity (STDP) (*Markram et al., 2012*).

In the STDP paradigm, coincident firing in the range of tens of milliseconds results in long-lasting changes in synaptic efficacy (*Feldman, 2012*). In classic STDP studies (*Bi and Poo, 1998*; *Feldman, 2000*; *Markram et al., 1997*; *Zhang et al., 1998*), presynaptic spikes leading postsynaptic spikes by a few milliseconds drive timing-dependent long-term potentiation (tLTP), whereas the opposite temporal ordering elicits timing-dependent tLTD. However, STDP is quite diverse, with rules and mechanisms depending on factors such as synapse type, developmental stage, and neuromodulation (*Debanne and Inglebert, 2023*; *Larsen and Sjöström, 2015*; *McFarlan et al., 2023*).

Many forms of STDP critically depend on NMDARs, which are a subfamily of glutamatergic receptors known for forming heterotetrameric ligand-gated ion channels (*Paoletti et al., 2013*; *Wong et al., 2021*). In the case of cortical tLTP, the role of NMDARs is well understood. Action potentials initiated at the soma are thought to backpropagate through dendrites (*Stuart and Sakmann, 1994*) to elicit nonlinear calcium signals localized to dendritic spines (*Koester and Sakmann, 1998*) by activating glutamate-bound NMDARs in the postsynapse (*Yuste and Denk, 1995*). These NMDARs are thus able to act as classic detectors of coincident EPSPs and action potentials in postsynaptic neurons (*Markram et al., 1997*) because of their dual need for presynaptically released glutamate as well as postsynaptic depolarization to relieve the $Mg^{2+}$ blockade and flux the $Ca^{2+}$ that triggers tLTP (*Sjöström and Nelson, 2002*; *Wong et al., 2021*).

In cortical tLTD, however, the role of NMDARs is not as well understood. We previously found evidence that, at L5 PC→PC connections, tLTD relies on preNMDAR signaling (*Sjöström et al., 2003*; *Sjöström et al., 2007*), but precisely how was not clear (*Duguid and Sjöström, 2006*). After all, the presynaptic spike is long gone by the time preNMDARs bind the released glutamate, arguing that another presynaptic spike must arrive soon enough to depolarize and unblock glutamate-bound preNMDARs, suggesting a need for high-frequency spiking (*Duguid and Sjöström, 2006*). Yet tLTD is readily induced at low frequencies (*Sjöström et al., 2003*).

More recently, we found that preNMDARs differentially regulate evoked and spontaneous release via RIM1aß and JNK2, respectively (*Abrahamsson et al., 2017*). Interestingly, the JNK-mediated regulation of spontaneous release by preNMDARs was not $Mg^{2+}$-sensitive (*Abrahamsson et al., 2017*), revealing the existence of an unconventional form of non-ionotropic NMDAR signaling in the axon (*Bouvier et al., 2018*; *Dore et al., 2017*; *Wong et al., 2021*). If tLTD relied on such non-ionotropic preNMDAR signaling, it would explain why it can be induced at both low and high frequencies, since in this signaling mode, preNMDARs are sensitive to glutamate but not to $Mg^{2+}$ or membrane potential.

Here, we explored how NMDARs signal in tLTD. We found that, regardless of frequency, L5 PC→PC tLTD relies on the non-ionotropic preNMDAR signaling pathway mediated by JNK2, which helps to resolve the long-standing enigma surrounding its lack of frequency dependence (*Duguid and Sjöström, 2006*).

## Results

### tLTD relies on presynaptically located NMDARs

We previously reported pharmacological and imaging evidence for preNMDARs at L5 PC→PC synapses that are necessary for neocortical tLTD (*Abrahamsson et al., 2017*; *Buchanan et al., 2012*; *Sjöström et al., 2003*; *Sjöström et al., 2007*), but their existence has been disputed (*Christie and Jahr, 2009*). We, therefore, wanted to directly challenge our previous findings by sparsely knocking

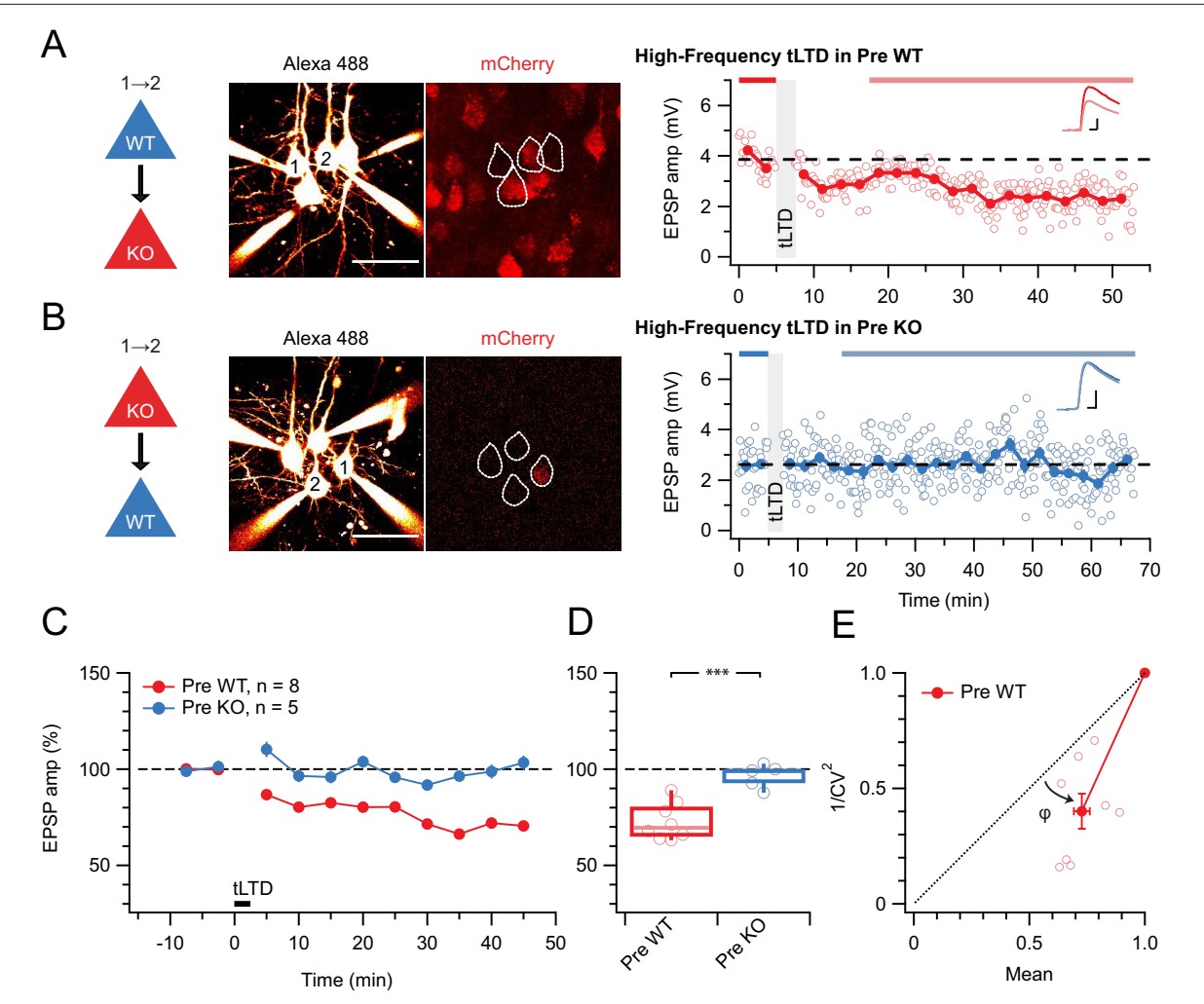

**Figure 1.** tLTD requires presynaptic NMDARs. (**A**) Sample quadruple recording with L5 PCs visualized with Alexa 488 (left, scale bar 50 μm) and NMDAR deletion indicated by mCherry (right). In this wild-type (WT) → KO sample, HF tLTD at 20 Hz (see Methods) was intact (after/before = 68%, p<0.001), suggesting postNMDARs were not required. Traces (right) were averaged over periods indicated by blue lines. Scale bars: 5 ms, 1 mV. (**B**) For this KO→WT sample, however, the same induction elicited no plasticity (after/before = 100%, p=0.99), suggesting tLTD needed preNMDARs. Scale bars as in A. (**C, D**) HF tLTD was robustly evoked for presynaptic WT (Pre WT, blue) but not presynaptic KO pairs (Pre KO, red), verifying the need for preNMDARs in tLTD that we previously demonstrated (***Sjöström et al., 2003***; ***Sjöström et al., 2007***). Pre WT: WT→WT and WT→KO pooled. Pre KO: KO→WT and KO→KO pooled. (**E**) CV analysis indicated that tLTD was presynaptically expressed ($\varphi$=19°±4°, n=8, one-sample t-test vs. diagonal, p0.001), in agreement with our prior studies (***Sjöström et al., 2003***; ***Sjöström et al., 2007***).

out (KO) the obligatory GluN1 NMDAR subunit (see Methods) and then measuring tLTD at synaptically connected L5 PC→PC pairs that lacked NMDARs pre- or postsynaptically.

With this approach, we found that preNMDAR deletion abolished high-frequency (HF) tLTD at 20 Hz (see Methods), whereas postsynaptic NMDAR (postNMDARs) deletion did not (***Figure 1***). We previously found that tLTD is expressed presynaptically (***Sjöström et al., 2003***; ***Sjöström et al., 2007***). We relied on the coefficient of variation (CV) analysis (***Brock et al., 2020***) to verify that postNMDAR deletion did not alter this (***Figure 1E***). We concluded that L5 PC→PC tLTD relied on pre- but not postNMDARs.

Throughout this study, boxplots show medians and quartiles, with whiskers denoting extremes, but data is reported as mean ± SEM, with n indicating the number of connections.

## tLTD does not rely on ionotropic NMDAR signaling

We previously imaged preNMDAR-mediated $Ca^{2+}$ supralinearities in axons (***Abrahamsson et al., 2017***; ***Buchanan et al., 2012***). We also found that $Mg^{2+}$-sensitive ionotropically signaling preNMDARs

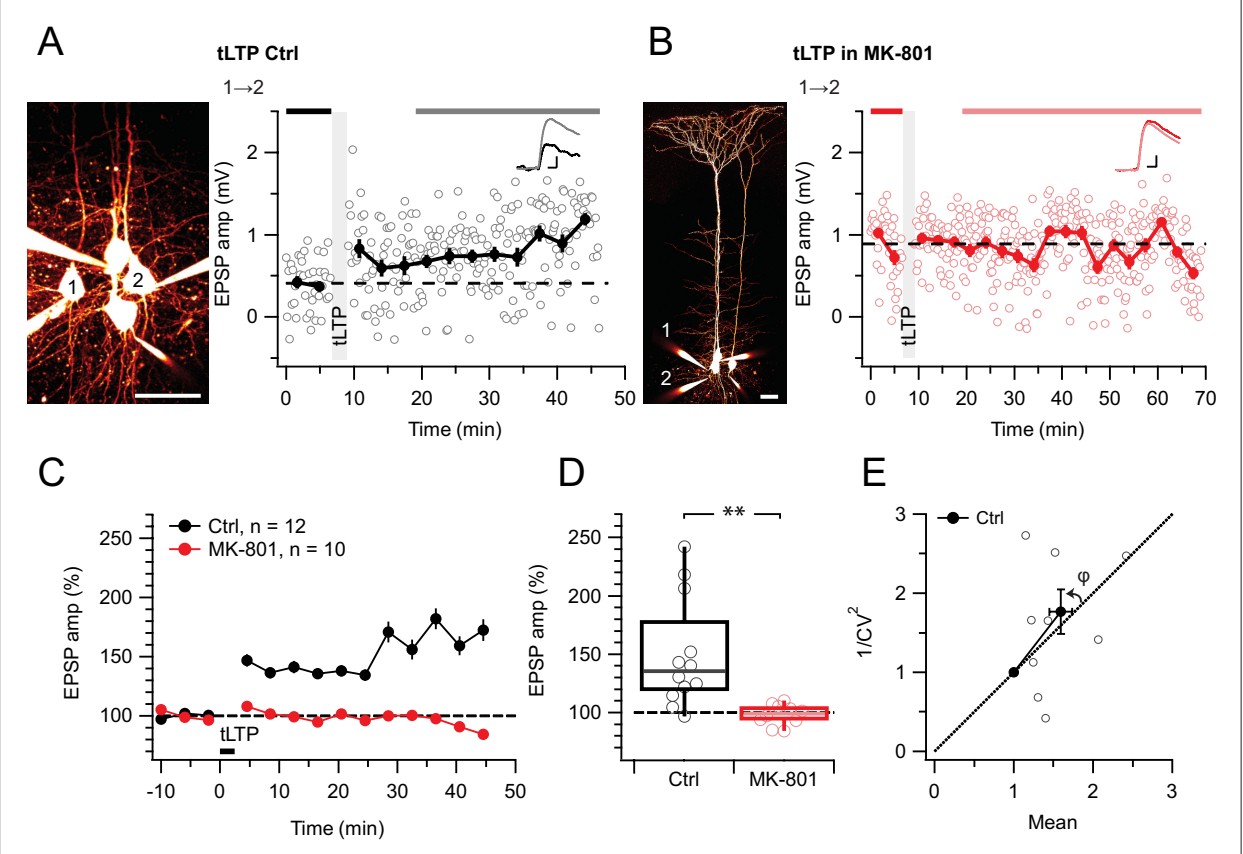

**Figure 2.** tLTP requires ionotropic NMDAR signaling. (**A**) At this sample connection (PC 1→2, see Alexa 594 fills at left, scale bar: 50 μm), tLTP induction was successful (after/before = 210%, p<0.001). Traces (right) were averaged over time periods indicated in black/gray. To avoid MK-801 affecting short-term depression (*Abrahamsson et al., 2017*; *Sjöström et al., 2003*), baseline spiking was 0.1 Hz. Scale bars: 5ms, 0.2 mV. (**B**) In this sample, MK-801 abolished tLTP with the same induction protocol (after/before = 95.44%, p=0.55). Baseline spiking and scale bars as in (**A**). (**C, D**) Ensemble data revealed that tLTP was robustly expressed in controls (black) but abolished in MK-801 (red). (**E**) CV analysis indicated that tLTP was expressed both pre- and postsynaptically to varying degrees across different pairs (φ=11°±15°, n=10, p=0.44, two data points without potentiation were excluded, see Methods), in agreement with our prior work (*Sjöström et al., 2007*).

The online version of this article includes the following figure supplement(s) for figure 2:

**Figure supplement 1.** Low-frequency evoked release did not respond to MK-801.

boost neurotransmitter release during periods of high-frequency activity at L5 PC→PC synapses and that this frequency dependence is inherited from the $Mg^{2+}$ blockade of the preNMDAR channel (*Abrahamsson et al., 2017*; *Wong et al., 2024*). Since tLTD is not frequency dependent (*Sjöström et al., 2003*), we hypothesized that tLTD is not sensitive to blockade of the NMDAR channel pore. To test this hypothesis, we relied on the NMDAR pore blocker MK-801 (*Song et al., 2018*).

We first wanted to establish a positive control. MK-801 is known to block hippocampal LTP (*Nabavi et al., 2013*) as well as tLTP at excitatory inputs onto neocortical L2/3 PCs (*Bender et al., 2006*; *Nevian and Sakmann, 2006*; *Rodríguez-Moreno et al., 2011*). At L5 PC→PC synapses, tLTP relies on different NMDARs than tLTD (*Sjöström et al., 2003*), but the MK-801 sensitivity has, to our knowledge, not been explored. Here, we found that MK-801 wash-in abolished potentiation (*Figure 2*), suggesting that L5 PC→PC tLTP relies on ionotropically signaling postNMDARs, like tLTP at other neocortical synapses (*Bender et al., 2006*; *Nevian and Sakmann, 2006*). We also verified that MK-801 wash-in had no effect on baseline responses elicited at low frequency (*Figure 2—figure supplement 1*), as expected (*Abrahamsson et al., 2017*; *Buchanan et al., 2012*; *Sjöström et al., 2003*; *Wong et al., 2024*).

With this positive control established, we next explored if L5 PC→PC tLTD and tLTP were equally sensitive to MK-801. However, MK-801 had no effect on tLTD (*Figure 3*). Since the action of MK-801 is activity and voltage-dependent (*Huettner and Bean, 1988*), we were concerned that HF tLTD and

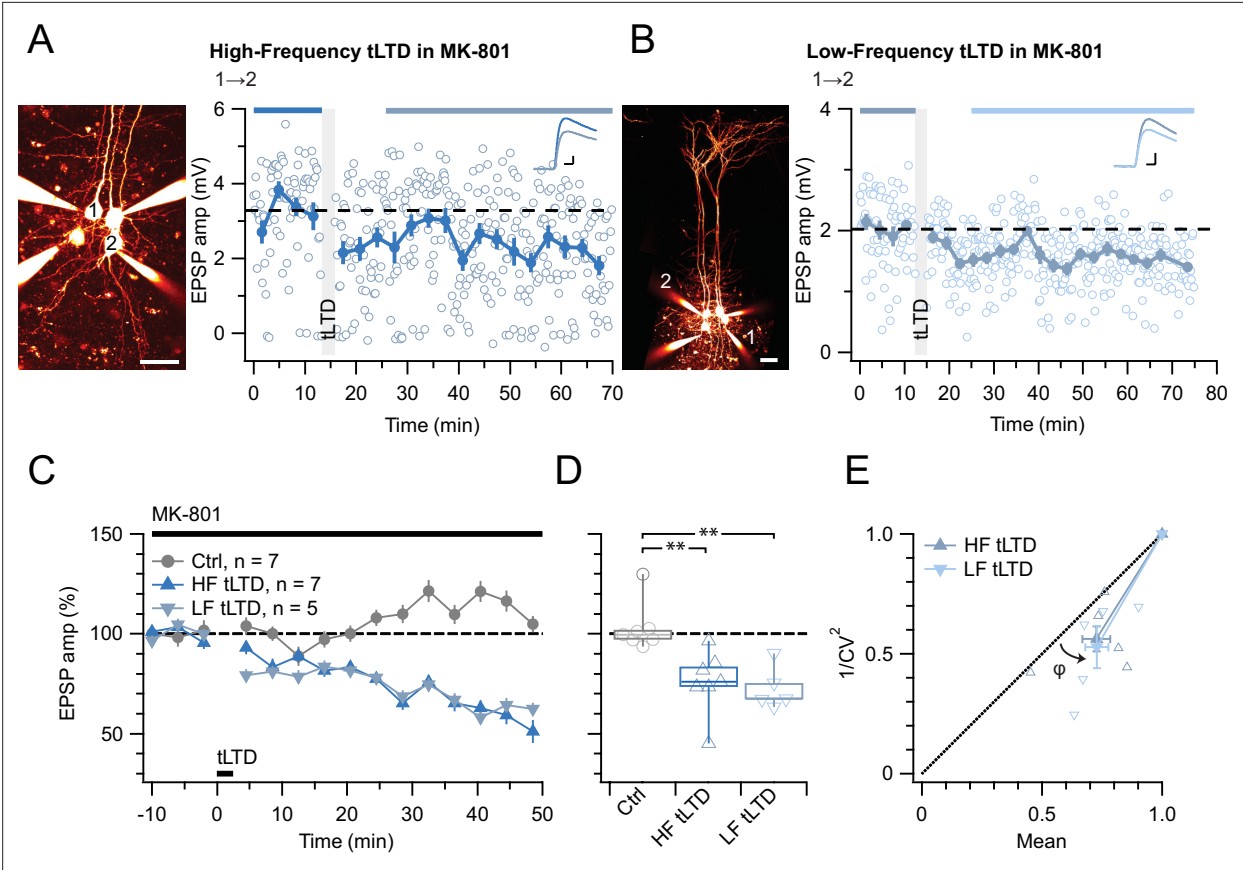

**Figure 3.** tLTD does not require ionotropic NMDAR signaling. (**A**) At this sample connection (PC 1→2, see Alexa 594 fills left, scale bar: 50 μm), HF tLTD was successfully induced in the presence of MK-801 (after/before = 74%, p<0.001). Traces were averaged over time periods indicated in blue/light blue. Baseline spiking was 0.1 Hz. Scale bars: 5 ms, 0.5 mV. (**B**) In this sample, LF tLTD at 1 Hz (see Methods) was successful in MK-801 (after/before = 78%, p<0.001). Baseline spiking and scale bars as in A. (**C, D**) Ensembles revealed robust tLTD at both low and high frequencies in MK-801 (blue triangles) compared to no-induction MK-801 controls (gray, Welch ANOVA p<0.01), suggesting that tLTD does not rely on ionotropic NMDAR signaling. (**E**) CV analysis indicated that tLTD at both induction frequencies was presynaptically expressed (HF LTD: φ=13°±5°, n=6, p<0.05; LF LTD: 15°±4°, n=5, p<0.05), in agreement with our prior findings (*Figure 1*; *Sjöström et al., 2003*; *Sjöström et al., 2007*). One experiment with <5% plasticity was excluded, see Methods.

The online version of this article includes the following figure supplement(s) for figure 3:

**Figure supplement 1.** Low-frequency evoked release was suppressed by 7-CK.

low-frequency (LF) tLTD (see Methods) might be differentially affected by this drug, but we found no difference (*Figure 3*).

Even with the positive tLTP control (*Figure 2*), we were concerned that the absence of effect of MK-801 on tLTD was a negative result (*Figure 3*). We, therefore, wished to strengthen our findings by complementing them with another approach. We thus attempted to block tLTD using 7-chlorokynurenate (7-CK), a co-agonist site blocker that abolishes NMDAR currents (*Nabavi et al., 2013*) without hindering glutamate binding (*Kemp et al., 1988*). In control experiments, we found that 7-CK suppressed synaptic responses (*Figure 3—figure supplement 1*) in a manner consistent with its known additional action as a competitive inhibitor of L-glutamate transport into synaptic vesicles (*Bartlett et al., 1998*). We, therefore, waited for synaptic responses to stabilize before inducing tLTD in 7-CK. With this approach, we found that 7-CK was unable to block tLTD (*Figure 4*).

A parsimonious interpretation of these experiments is that tLTP but not tLTD requires ionotropic NMDAR signaling. This interpretation additionally helps to explain why tLTD is not frequency dependent (*Sjöström et al., 2003*).

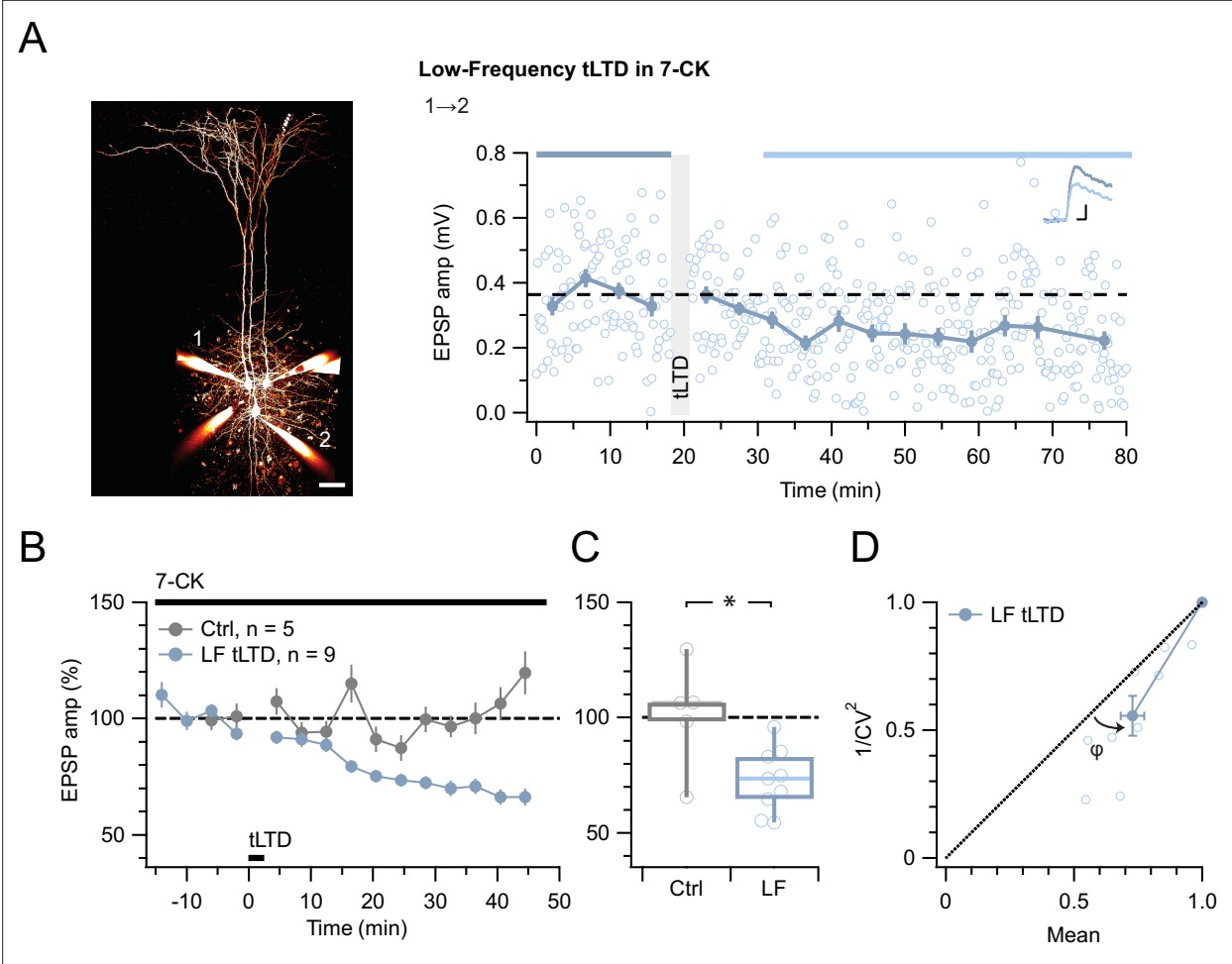

**Figure 4.** tLTD does not require ligand binding at NMDAR co-agonist site. (**A**) At this sample connection (PC 1→2, see Alexa 594 fills at left, scale bar 50 μm), LF tLTD at 1 Hz (see Methods) persisted in the presence of 7-CK (after/before = 68 %, p<0.001). Traces were averaged over time periods indicated in blue/light blue. Baseline spiking was 0.1 Hz. Scale bars: 5 ms, 0.1 mV. (**B, C**) Ensemble revealed that LF tLTD persisted in 7-CK (blue) compared to no-induction 7-CK controls (gray), verifying that it does not require ionotropic NMDAR signaling. (**D**) CV analysis verified that LF tLTD was presynaptically expressed ($\varphi$=14°±3°, n=9, p<0.01), in agreement with our prior findings (*Figures 1 and 3E*; *Sjöström et al., 2003*; *Sjöström et al., 2007*).

## tLTD relies on JNK2 but not on RIM1αβ

We previously found that ionotropic preNMDAR-mediated regulation of evoked release at L5 PC→PC synapses relies on RIM1αβ (*Abrahamsson et al., 2017*). Additionally, RIM1α is required for endocannabinoid-mediated LTD in hippocampus (*Castillo et al., 2002*; *Chevaleyre et al., 2007*) as well as for LTP in amygdala (*Fourcaudot et al., 2008*). We, therefore, explored if RIM1αβ signaling similarly contributed to L5 PC→PC tLTD, by conditionally deleting RIM1αβ in excitatory neurons (see Methods and *Abrahamsson et al., 2017*).

We found that L5 PC→PC tLTD was robust in Emx1$^{Cre/+}$; RIM1αβ$^{fl/fl}$ animals (*Figure 5C–E*). The magnitude of tLTD in Emx1$^{Cre/+}$; RIM1αβ$^{fl/fl}$ mice was furthermore indistinguishable to that in WT mice (*Figure 5C and D*). To explore the locus of expression of tLTD, we analyzed paired-pulse ratio (PPR) and CV, which both suggested a presynaptic site of expression (*Figure 5E and F*), like we found before (*Figures 1, 3E and 4D*; *Sjöström et al., 2003*; *Sjöström et al., 2007*). Homozygous RIM1αβ deletion thus had no detectable effect on L5 PC→PC tLTD.

Since we previously found that non-ionotropic preNMDAR signaling requires JNK2 to control spontaneous release (*Abrahamsson et al., 2017*), we were curious to test the potential need for JNK2 in tLTD. We blocked JNK2 signaling by pre-incubating (see Methods) with the specific inhibitor

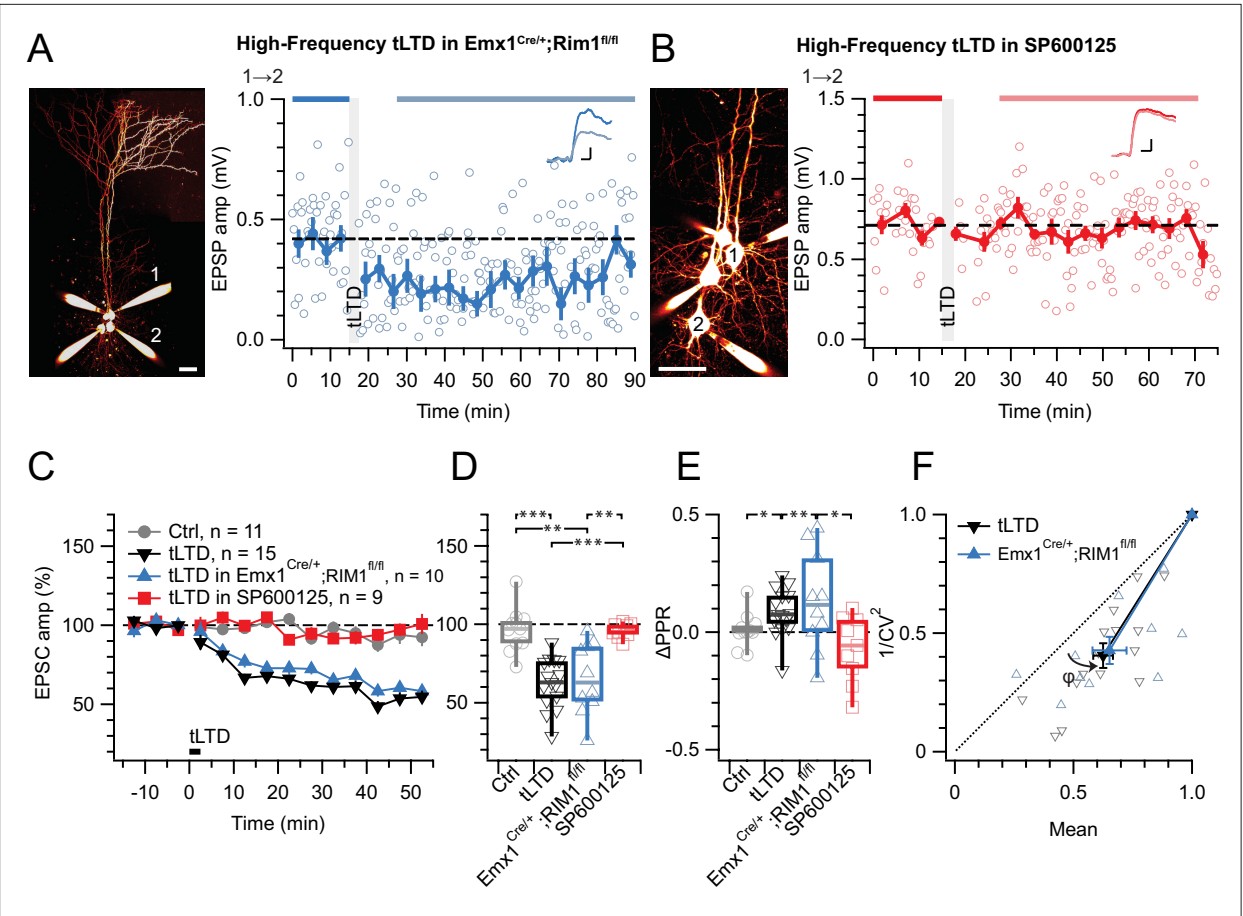

**Figure 5.** tLTD does not require RIM1αβ but relies on JNK2. (**A**) At this sample PC 1→2 connection (left: Alexa 594 fills, scale bar 50 µm) in an acute slice from a homozygous RIM1αβ deletion mouse, tLTD persisted (after/before = 59%, p<0.001). Scale bars: 5ms, 0.1 mV. (**B**) For this sample connection, the JNK2-blocker SP600125 (**Bennett et al., 2001**) abolished tLTD (after/before = 99%, p=0.86). Scale bars: 5ms, 0.2 mV. (**C, D**) Ensemble data revealed that tLTD after homozygous RIM1αβ deletion (blue) was indistinguishable from WT tLTD (black), thereby dissociating tLTD from RIM1αβ. However, SP600125 robustly disrupted tLTD (red), with an outcome indistinguishable from no-induction controls (gray). Welch ANOVA p<0.001. (**E**) tLTD increased PPR (black and blue) compared to controls (gray), suggesting presynaptic expression. Welch ANOVA p<0.05. (**F**) CV analysis indicated that tLTD was presynaptically expressed, whether RIM1αβ was deleted ($\varphi$=16°±4°, n=10, p<0.01) or not ($\varphi$=13°±2°, n=15, p<0.001), in agreement with our prior findings (**Figures 1 and 3E**, **Figure 4D**; **Sjöström et al., 2003**; **Sjöström et al., 2007**).

SP600125 (**Bennett et al., 2001**), which abolished tLTD (**Figure 5B–E**). Taken together, our results show that tLTD relies on JNK2 but not on RIM1αβ.

## tLTD relies on presynaptic JNK2 signaling regardless of frequency

To selectively disrupt JNK2/STX1a interactions and associated signaling, we loaded a cell-impermeable variant of the JGRi1 peptide (**Marcelli et al., 2019**) into pre- or postsynaptic PCs via the patch pipettes. When loaded presynaptically, HF tLTD was abolished, whereas when loaded postsynaptically, HF tLTD could be induced (**Figure 6A–D**). As before, tLTD was expressed presynaptically (**Figure 6E and F**). This suggested that tLTD was mediated by preNMDAR signaling via the JNK2/STX1a complex.

By comparison with preNMDAR regulation of spontaneous release via JNK2 — which was independent of $Mg^{2+}$ and frequency (**Abrahamsson et al., 2017**) — this also suggested that the JGRi1 peptide ought to block tLTD at any frequency. We, therefore, repeated these JGRi1 peptide loading experiments for LF tLTD. As for HF tLTD, when the peptide was loaded presynaptically, LF tLTD was abolished, whereas when loaded postsynaptically, LF tLTD was expressed by presynaptic downregulation of release (**Figure 6G–L**). We note that at low frequency, preNMDARs remained blocked by $Mg^{2+}$ (**Abrahamsson et al., 2017**; **Wong et al., 2024**), so this outcome lends further support to the

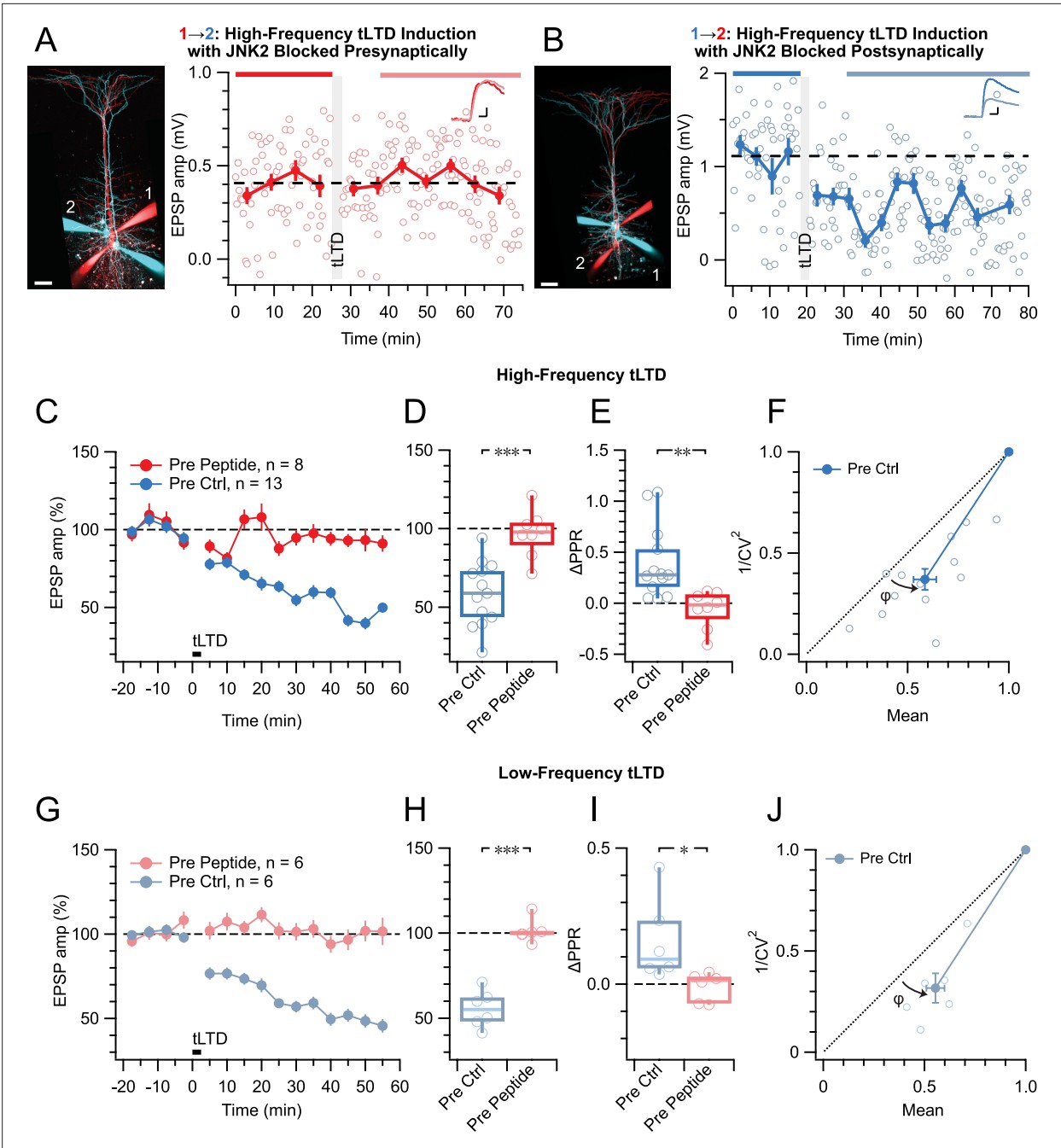

**Figure 6.** tLTD relies on presynaptic JNK2/STX1 signaling. (**A**) Presynaptic loading of peptide disrupting JNK2/STX1 interactions (*Marcelli et al., 2019*) in this PC 1→2 pair abolished HF tLTD (after/before = 105%, p=0.52). Red cells were loaded with peptide, whereas blue cells were not. Traces were averaged over time periods indicated in red/pink. Scale bar 50 µm, inset scale bars 5 ms, 0.1 mV. (**B**) Postsynaptic peptide loading in this PC 1→2 pair had no effect on HF tLTD (after/before = 47%, p<0.001). Inset scale bars 5ms, 0.2 mV. (**C, D**) Presynaptic (red) but not postsynaptic/no peptide loading (blue) abolished HF tLTD, demonstrating a need for presynaptic JNK2/STX1 signaling. (**E**) Without presynaptic peptide (blue), HF tLTD induction increased PPR as expected from reduced release, whereas with presynaptic peptide (red), PPR expectedly remained unaffected. (**F**) CV analysis of HF tLTD with no presynaptic peptide agreed with presynaptic expression ($\varphi$=13°±3°, n=13, p<0.001). (**G, H**) Presynaptic (red) but not postsynaptic/no peptide loading (blue) abolished LF tLTD, demonstrating that the need for presynaptic JNK2/STX1 signaling was not frequency dependent like ionotropic preNMDAR signaling is (*Abrahamsson et al., 2017*). (**I**) Without presynaptic peptide (blue), LF tLTD induction increased PPR in keeping with presynaptic expression, but with presynaptic peptide (red), PPR was unaffected. (**J**) CV analysis of LF tLTD with no presynaptic peptide agreed that release was reduced ($\varphi$=11°±2°, n=6, p<0.01).

view that tLTD requires non-ionotropic signaling. The sensitivity to pre- but not postsynaptic peptide further cements the view that this form of NMDAR signaling is located presynaptically.

Taken together, these results demonstrate that, regardless of induction frequency, tLTD is mediated by non-ionotropic preNMDAR signaling via the JNK2/STX1a complex in presynaptic neurons.

## Discussion

The role of preNMDARs in tLTD has long been enigmatic (*Duguid and Sjöström, 2006*; *Sjöström et al., 2003*). Our study provides resolution to this long-standing enigma by showing for the first time that tLTD relies on non-ionotropic preNMDAR signaling via JNK2. Our findings thereby further challenge the traditional view of NMDA receptors as ionotropic coincidence detectors and reveal how non-ionotropic preNMDAR signaling can shape STDP.

### Reconciling decades of preNMDAR controversy

Although evidence for preNMDARs has been reported for decades (*Aoki et al., 1994*; *Berretta and Jones, 1996*; *DeBiasi et al., 1996*; *Glitsch and Marty, 1999*; *Liu et al., 1997*), this receptor type has been controversial. For instance, one study (*Christie and Jahr, 2009*) could not find the preNMDARs that we reported as underlying L5 PC→PC tLTD (*Sjöström et al., 2003*; *Sjöström et al., 2007*). Several other studies, however, largely corroborated our findings of preNMDARs at L4 PC → L2/3 PC synapses and elsewhere (*Banerjee et al., 2014*; *Banerjee et al., 2009*; *Bender et al., 2006*; *Brasier and Feldman, 2008*; *Corlew et al., 2007*; *Rodríguez-Moreno and Paulsen, 2008*). Then again, one study argued that the NMDARs that underlie L4 PC → L2/3 PC tLTD are actually postsynaptic (*Carter and Jahr, 2016*). Similar disagreements exist in cerebellum, with some studies reporting preNMDARs (*Casado et al., 2000*; *Casado et al., 2002*; *Glitsch and Marty, 1999*) and others arguing that they do not exist (*Christie and Jahr, 2008*).

Central to the preNMDAR controversy has been how they signal. Classically, NMDARs need both glutamate and depolarization to relieve $Mg^{2+}$ blockade, open up, and signal via calcium flux (*Dore et al., 2017*; *Wong et al., 2021*). Consistently, preNMDARs often require high-frequency spike trains to activate (*Abrahamsson et al., 2017*; *Bidoret et al., 2009*; *Buchanan et al., 2012*; *McGuinness et al., 2010*; *Sjöström et al., 2003*; *Wong et al., 2021*), as a single spike that elicits glutamate release is gone by the time preNMDARs bind the glutamate. However, preNMDARs also govern spontaneous release (*Abrahamsson et al., 2017*; *Berretta and Jones, 1996*; *Buchanan et al., 2012*; *Kunz et al., 2013*; *Sjöström et al., 2003*) and tLTD (*Sjöström et al., 2003*), which both occur at low frequencies, leading to a long-standing contradiction in the field (*Banerjee et al., 2009*; *Duguid and Sjöström, 2006*; *Sjöström et al., 2003*).

This controversy can be resolved by several intriguing preNMDAR properties. Whereas postN-MDARs are ubiquitously expressed, preNMDARs are only found at specific synapse types (*Banerjee et al., 2014*; *Brasier and Feldman, 2008*; *Buchanan et al., 2012*; *Larsen et al., 2014*). PreNMDARs are thus easy to miss if different synapse types are mixed experimentally. Furthermore, preNMDARs often rely on particular subunits and conditions that reduce the flux of $Ca^{2+}$ (*Banerjee et al., 2009*; *Kunz et al., 2013*; *Larsen et al., 2011*), leading to weak signals that may be hard to detect.

Finally, preNMDARs can also signal non-ionotropically to control release (*Abrahamsson et al., 2017*; *Wong et al., 2024*). If the expectation is that NMDARs can only signal ionotropically, this may confound experimental interpretation. For instance, manipulations such as loading cells internally with MK-801 to block NMDARs from the inside or with BAPTA to chelate $Ca^{2+}$ may become inconclusive. However, non-ionotropic NMDAR signaling has been reported broadly, for instance, in classic LTD (*Nabavi et al., 2013*; *Nielsen et al., 2024*; *Park et al., 2022*), structural plasticity (*Park et al., 2022*; *Stein et al., 2021*; *Thomazeau et al., 2021*), and ischemic excitotoxicity (*Weilinger et al., 2016*), so this mode of NMDAR signaling is not necessarily rare.

In fact, non-ionotropic signaling has also been found for other ionotropic channels, such as kainate receptors (*Rodríguez-Moreno and Lerma, 1998*), AMPA receptors (*Takago et al., 2005*; *Wang and Durkin, 1995*), and even voltage-gated $Ca^{2+}$ channels (*Trus and Atlas, 2024*). The classic textbook view of receptor types as strictly ionotropic or non-ionotropic is, therefore, not solidly anchored in the biology.

Our finding that preNMDAR signaling in L5 PC→PC tLTD is non-ionotropic explains why this form of plasticity does not depend on frequency (*Duguid and Sjöström, 2006*; *Sjöström et al., 2003*). This is because, in this signaling mode, preNMDARs are sensitive to glutamate but not to Mg²⁺ or membrane potential, so there is no need for spikes to arrive in quick succession to provide the depolarization that would be required for ionotropic preNMDAR signaling (*Duguid and Sjöström, 2006*).

Overall, our finding that non-ionotropically signaling preNMDARs are key to L5 PC→PC tLTD thus aligns with much of the prior literature while simultaneously bringing closure to long-standing controversies (*Bouvier et al., 2018*). Our genetic deletion approach furthermore firmly situates the NMDARs needed for tLTD in the pre- rather than the postsynaptic cell of L5 PC→PC pairs (*Carter and Jahr, 2016*).

## A hitherto unappreciated role for JNK2 in tLTD

JNKs are serine-threonine kinases that belong to the mitogen-activated protein kinase family and mediate stress stimuli, e.g., cytokines, ultraviolet irradiation, and heat shock. There are three closely related vertebrate genes: JNK1 and JNK2 are ubiquitous, while JNK3 is primarily neuronal (*Yamasaki et al., 2012*). In the brain, JNKs are key developmental regulators, for instance, of neuronal migration, dendrite formation, and axon maintenance (*Yamasaki et al., 2012*).

We previously demonstrated that tLTD at L5 PC→PC connections requires simultaneous activation of preNMDARs and endocannabinoid CB1 receptors (*Sjöström et al., 2003*). It has long been known that JNK is activated by NMDARs (*Coffey, 2014*; *Mukherjee et al., 1999*) as well as by CB1 receptors (*Rueda et al., 2000*). More recently, we and others established that preNMDARs regulate spontaneous release by signaling via JNK2 (*Abrahamsson et al., 2017*; *Nisticò et al., 2015*) and that this regulation critically depends on an interaction between JNK2 and STX1a (*Marcelli et al., 2019*). Interestingly, it was previously proposed that JNK2 signaling is involved in behavioral learning as well as in classical NMDAR-dependent LTD (*Curran et al., 2003*; *Morel et al., 2018*). Our study, however, is to our knowledge the first to show a need for JNK2 signaling specifically in tLTD, which relies on mechanistic underpinnings that differ from classical LTD, such as retrograde endocannabinoid signaling (*Bender et al., 2006*; *Nevian and Sakmann, 2006*; *Sjöström et al., 2003*). Furthermore, in our hands, the activation of JNK2 is achieved by non-ionotropic preNMDAR action. Our present study thus extends the prior literature by suggesting that JNK2 signaling may be a general principle that applies to distinct forms of long-term plasticity.

## Diverse preNMDAR signaling across different synapse types

In our hands, the NMDAR channel pore blocker MK-801 surprisingly did not block tLTD at L5 PC→PC connections, even though several other studies of tLTD at L4 PC → L2/3 PC synapses reported that MK-801 abolishes tLTD (*Banerjee et al., 2014*; *Larsen et al., 2014*; *Rodríguez-Moreno et al., 2013*; *Rodríguez-Moreno et al., 2011*; *Rodríguez-Moreno and Paulsen, 2008*). This difference is likely due to non-ionotropic versus ionotropic preNMDAR tLTD at L5 PC→PC and L4 PC → L2/3 PC synapses, respectively. This mechanistic distinction is generally consistent with the view that STDP depends on synapse type (*Larsen and Sjöström, 2015*; *McFarlan et al., 2023*). In fact, tLTD is mediated by distinct mechanisms even for different synaptic input types onto the same L2/3 PCs (*Banerjee et al., 2009*; *Larsen et al., 2014*).

A corollary from this synapse-specific difference in non-ionotropic versus ionotropic preNMDAR tLTD at L5 PC→PC and L4 PC → L2/3 PC synapses is a differential frequency dependence. Indeed, at L4 PC → L2/3 PC synapses, a frequency-dependent form of presynaptic self-depression has been reported (*Rodríguez-Moreno et al., 2013*). Such frequency dependence of tLTD does not, however, exist at L5 PC→PC connections (*Sjöström et al., 2003*).

Curiously, preNMDARs at L5 PC→PC connections signal ionotropically to boost neurotransmitter release during high-frequency firing (*Abrahamsson et al., 2017*; *Buchanan et al., 2012*; *Sjöström et al., 2003*; *Wong et al., 2024*). This boosting relies on RIM1αβ (*Abrahamsson et al., 2017*) and mTOR-mediated protein synthesis in L5 PC axons (*Wong et al., 2024*), yet is synapse-type-specific, so does not affect L5 PC→interneuron synapses (*Buchanan et al., 2012*; *Wong et al., 2024*). Consequently, MK-801 blocks preNMDAR-mediated boosting of evoked L5 PC→PC release (*Abrahamsson et al., 2017*; *Buchanan et al., 2012*; *Sjöström et al., 2003*).

Another corollary from non-ionotropic preNMDAR signaling in tLTD is that removing $Mg^{2+}$ to unblock the channel pore should not induce LTD at L5 PC→PC synapses during low-frequency firing. In agreement, reduced $Mg^{2+}$ does not elicit LTD but actually boosts neurotransmission (*Abrahamsson et al., 2017*; *Wong et al., 2024*). Whether or not $Mg^{2+}$ washout promotes LTD at L4 PC → L2/3 synapses has not been explored, as far as we know.

### Caveats

We relied on JNK2 blockade as a proxy for non-ionotropic preNMDAR signaling, as we established this hallmark feature of flux-independent preNMDAR signaling of L5 PCs in a previous study (*Abrahamsson et al., 2017*). It is, however, presently unclear how well this proxy generalizes. A drug that specifically blocks non-ionotropic but not ionotropic NMDAR signaling would resolve this, much like MK-801 does the vice versa. To our knowledge, such a drug presently does not exist.

As mentioned above, PreNMDAR and CB1 receptor co-activation is required for L5 PC→PC tLTD (*Sjöström et al., 2003*). Although L5 PC→PC tLTD does not depend on frequency (*Sjöström et al., 2003*), consistent with the non-ionotropic preNMDAR signaling we provide evidence for here, chemical LTD (cLTD) induced by wash-in of cannabinoid CB1 receptor antagonists is, in fact, frequency dependent (*Sjöström et al., 2003*). This apparent discrepancy is difficult to explain, because tLTD requires CB1 receptor signaling at low as well as high frequencies (*Sjöström et al., 2003*). Although the frequency dependence of cLTD is consistent with its known sensitivity to MK-801 (*Sjöström et al., 2003*), an apparent disagreement remains. A potential explanation is that CB1 receptor signaling in tLTD is phasic, whereas that in cLTD is tonic, and phasic/tonic CB1 signaling paths are known to differ mechanistically (*Castillo et al., 2012*). To address this, rapid photolysis of caged CB1 receptor agonists (*Heinbockel et al., 2005*) may mimic tLTD better than slow wash-in of cannabinoid CB1 receptor antagonists (*Sjöström et al., 2003*). Resolving this enigma will require future work.

NMDAR signaling that does not involve ion flux has sometimes been termed *metabotropic* (e.g. *Nabavi et al., 2013*). This designation can be confusing, as it could imply G-protein coupling (*Heuss and Gerber, 2000*), which has not been conclusively demonstrated for NMDARs. We, therefore, prefer to use the term *non-ionotropic* until future studies reveal if the underlying mechanism is indeed G-protein linked.

Our experiments were conducted in juvenile visual cortex, so caution is warranted when extrapolating our findings to mature circuits, which may rely on different plasticity mechanisms (*Banerjee et al., 2009*; *Corlew et al., 2007*; *Larsen et al., 2014*; *Martínez-Gallego et al., 2022*). Further studies in older animals will be important to determine whether the adult brain relies on non-ionotropic preNMDAR signaling in STDP.

### Future directions and implications for disease

Our study provides a fresh perspective on non-ionotropic function for preNMDARs in tLTD at neocortical synapses. By engaging JNK2-mediated signaling independent of $Ca^{2+}$ influx, preNMDARs contribute to STDP in a manner that has not been previously appreciated. Our findings challenge the traditional view of NMDARs as coincidence detectors in Hebbian plasticity and highlight the diversity of synaptic plasticity mechanisms (*Larsen and Sjöström, 2015*; *McFarlan et al., 2023*). Future studies will be needed to explore the broader implications of non-ionotropic NMDAR signaling in other brain regions and under different physiological conditions.

It is crucial to understand how NMDARs operate, as they are hotspots for major synaptic pathologies such as Alzheimer, schizophrenia, and epilepsy (*Nugent et al., 2022*; *Paoletti et al., 2013*). For instance, a central hypothesis in schizophrenia research is based on NMDAR hypofunction (*Lisman et al., 2008*). Yet, if one were to rationally create an NMDAR-based therapy for schizophrenia, one would need to know which type of NMDAR signaling is relevant. Future studies may thus reveal the potential therapeutic relevance of targeting distinct NMDARs signaling pathways.

## Methods

### Animals and ethics statement

The animal study was reviewed as Animal Use Protocol (AUP) 6041 and approved by the Montreal General Hospital Facility Animal Care Committee (The MGH FACC) and adhered to the guidelines

of the Canadian Council on Animal Care (CCAC). At postnatal days 11–18, male or female mice were anaesthetized with isoflurane and sacrificed once the hind-limb withdrawal reflex was lost. Transgenic animals had no abnormal phenotype. Sparse NMDAR deletion was achieved by removing the obligatory GluN1 subunit in a subset of L5 PCs by neonatal injection (*Kim et al., 2014*) of AAV-eSYN-mCherry-iCre into V1 of Grin1$^{fl/fl}$ mice (a.k.a. NR1$^{flox}$) obtained from The Jackson Laboratory (#005246, see below), thus achieving sparse and conditional KO of the *Grin1* gene in these mice. A Cre-loxP recombinase strategy (*Nagy, 2000*) was used to generate transgenic mice after two generations with the *Rims1* gene homozygously conditionally deleted in excitatory cells, as genome-wide *Rims1* KO impairs survival (*Mittelstaedt et al., 2010*). Homozygous Emx1$^{Cre/Cre}$ mice (*Gorski et al., 2002*) were obtained from The Jackson Laboratory (#005628). Homozygous RIM1αβ$^{fl/fl}$ mice (*Kaeser et al., 2008*) were kindly gifted by Pascal Kaeser (Harvard University, MA). Heterozygous Emx1$^{Cre/+}$; RIM1αβ$^{fl/+}$ mice were generated by crossing Emx1$^{Cre/Cre}$ with RIM1αβ$^{fl/fl}$ mice. Emx1$^{Cre/+}$; RIM1αβ$^{+/+}$ and RIM1αβ$^{fl/fl}$ mice were generated by crossing Emx1$^{Cre/Cre}$; RIM1αβ$^{fl/+}$ mice with RIM1αβ$^{fl/+}$; no-Cre mice (*Abrahamsson et al., 2017*). These were distributed in a Mendelian fashion and had viability indistinguishable from that of C57BL/6 mice. To determine the genotype of each animal, tail biopsy and tattooing were performed on mice before the age of P6. Genotyping was carried out using standard methodology with Jackson Laboratory primers (RIM1: 12061, 12062; Emx1: oIMR1084, oIMR1085, oIMR4170, oIMR4171) using QIAGEN HotStarTaq DNA Polymerase kit (203203) and dNTPs from Invitrogen/Thermo Fisher (18427–013) (*Abrahamsson et al., 2017*). WT denotes C57BL/6 J (Jackson Laboratory #000664), RIM1αβ$^{fl/fl}$; no-Cre and genetically unaffected littermates (Emx1$^{Cre/+}$; RIM1αβ$^{+/+}$).

## Viral injections

Cre recombinase was delivered by viral injection of AAV9-eSYN-mCherry-T2A-iCre-WPRE (Vector Biolabs, Cat No. VB4856) into the primary visual cortex of Grin1$^{fl/fl}$ neonates (P0-2) to generate a conditional NMDAR deletion through Cre-loxP recombination at the site of the Grin1 gene. This cuts the Grin1 genetic sequence, thus preventing the cell from producing the GluN1 subunit. As the GluN1 subunit is obligate, the cells expressing the virus will not express functional NMDARs. Cells expressing the viral construct were detected via an mCherry tag. Pups were anesthetized by putting them on ice and viral injection was delivered with a needle syringe held in a stereotactic injection setup and connected to a microinjector apparatus. The animal head was held in place with ear bars and the tip of the injection needle was zeroed to lambda. The needle was then positioned to the following coordinates: x = ±1.10; y=0.00. The needle was lowered until it reached the pial surface, where the z coordinate was zeroed. Three injections of 0.2–0.3 µl each were performed at z1=–0.20; z2=–0.15; and z3=–0.10. Both hemispheres were injected to increase the number of slices available for experiments and to reduce the risk of seeing no expression in an animal on the experimental day, both of which enhanced productivity. The AAV9 serotype has a particularly high tropism for the central nervous system (*Foust et al., 2009*) and the enhanced synapsin (eSYN) promoter specifically targets neurons (*Hioki et al., 2007*). T2A is a self-cleaving peptide and facilitates co-expression of Cre recombinase and mCherry (*Liu et al., 2017*). Finally, the Woodchuck Hepatitis Virus (WHP) Post-transcriptional Regulatory Element (WPRE) enhances expression levels of the viral-encoded proteins (*Klein et al., 2006*), allowing successful expression in neocortical neurons, including pyramidal cells. By controlling the viral titer, expression levels were regulated to achieve sparse genetic deletion of NMDARs in primary visual cortex neurons (*Kim et al., 2014*).

## Acute slice preparation

After decapitation, the brain was removed and placed in ice-cold (~4 °C) artificial cerebrospinal fluid (ACSF), containing in mM: 125 NaCl, 2.5 KCl, 1 MgCl$_2$, 1.25 NaH$_2$PO$_4$, 2 CaCl$_2$, 26 NaHCO$_3$, and 25 glucose, bubbled with 95% O$_2$/5% CO$_2$. Osmolarity of the ACSF was adjusted to 338 mOsm with glucose. Oblique coronal 300-µm-thick acute brain slices were prepared using a Campden Instruments 5000 mz-2 vibratome (Lafayette Instrument, Lafayette, IN, USA). Brain slices were kept at ~33 °C in oxygenated ACSF for ~15 min and then allowed to cool at room temperature for at least 1 hr before we started the recordings.

## Electrophysiology

Experiments were carried out with ACSF heated to 32–34°C with a resistive inline heater (Scientifica Ltd), with temperature recorded and verified offline. If outside this range, recordings were truncated or not used. Patch pipettes with a resistance of 4–6 MΩ were pulled using a P-1000 electrode puller (Sutter Instruments, Novato, CA, USA) from medium-wall capillaries. Pipettes were filled with K-gluconate internal solution containing in mM: KCl, 5; K-Gluconate, 115; K-HEPES, 10; MgATP, 4; NaGTP, 0.3; Na-Phosphocreatine, 10; and 0.1% w/v Biocytin, adjusted with KOH to pH 7.2–7.4 and sucrose to osmolality of 310 mOsm (*Abrahamsson et al., 2017*). 40 µM and 80 µM Alexa Fluor 594 or Alexa Fluor 488 dyes, respectively, were supplemented to internal solution to permit post-hoc analysis of cell morphology with two-photon laser-scanning microscopy (*Blackman et al., 2014*; *Lalanne et al., 2016*). Neurons were patched using infrared video Dodt contrast (built in-house with Thorlabs equipment) with an Olympus LUMPlanFL N ×40/0.80 objective on a customized microscope (SliceScope, Scientifica Ltd, UK). Primary visual cortex (V1) was distinguished from surrounding V2 based on the presence of cortical layer 4. PCs in L5 of V1 were targeted based upon their prominent apical dendrite and large triangular somata. Morphometry and cell identity were verified using 2-photon microscopy of Alexa 594/488 fluorescence. Whole-cell recordings were carried out using BVC-700A amplifiers (Dagan Corporation, Minneapolis, MN). Recordings in current-clamp mode were acquired at 40 kHz with PCI-6229 boards (National Instruments, Austin, TX) using our in-house MultiPatch software (*Watanabe et al., 2023*) (https://github.com/pj-sjostrom/MultiPatch) (*Sjöström, 2025*) running in Igor Pro 7–9 (WaveMetrics Inc, Lake Oswego, OR).

Since the rate of connectivity in rodent primary visual cortex among L5 PCs is only 10–15% (*Sjöström et al., 2001*; *Song et al., 2005*), quadruple whole-cell recording was employed to enable the simultaneous testing of 12 PC→PC connections (*Abrahamsson et al., 2016*). GΩ seals were first formed on all four cells followed by rapid successive breakthrough, to avoid plasticity washout (*Lalanne et al., 2016*). To identify monosynaptic PC→PC connections in current-clamp mode, five spikes were evoked in each cell at 30 Hz by 5-ms-long current injections of 1.3-nA magnitude, repeated every 20 s, and averaged across 10 repetitions. A 250-ms-long test pulse of –25 pA was used to measure input and series resistance. Spike trains in different cells were separated from one another by 700ms to prevent accidental induction of long-term plasticity (*Abrahamsson et al., 2017*; *Lalanne et al., 2016*; *Sjöström et al., 2003*). If no PC→PC connections were detected, all four recordings were stopped, and another set of four cells was patched with new pipettes, either nearby or in a new acute slice.

In paired recordings, synaptic responses were strictly unitary and subthreshold (*Chou et al., 2025*; *Song et al., 2005*), ensuring that inhibitory circuits were not inadvertently recruited. There was, therefore, no need to pharmacologically block GABAergic transmission.

## STDP Experiments

We induced tLTD by evoking postsynaptic firing 25ms before presynaptic firing, either at 20 Hz or at 1 Hz (*Sjöström et al., 2001*; *Sjöström et al., 2003*), which we refer to as HF and LF tLTD, respectively, throughout the present study. tLTP was induced by evoking presynaptic firing 10ms before postsynaptic cell firing at 50 Hz (*Sjöström et al., 2001*).

To ensure a good signal-to-noise ratio, we only used PC→PC connections >0.3 mV, like before (*Sjöström et al., 2007*). For *Figure 1* through 4, presynaptic cells were spiked once every 10 s (see below for clarification). For *Figures 5 and 6*, presynaptic cells were typically spiked five times at 30 Hz every 20 s during baseline periods, resulting in trains of EPSPs, which we denote $EPSP_1$ through $EPSP_5$. We assessed plasticity by calculating the ratio of $EPSP_1$ amplitude averaged across post-induction and pre-induction periods, as indicated in figures. Synaptic stability was assessed using a *t*-test of Pearson's *r* for $EPSP_1$ across the pre-induction baseline period, with p<0.05 implying instability. Recordings with unstable baseline, >30% change in input resistance, or >8 mV change in resting membrane potential were truncated or discarded, like before (*Sjöström et al., 2001*; *Sjöström et al., 2003*; *Sjöström et al., 2007*).

To determine the locus of neocortical STDP expression, analysis of the CV of $EPSP_1$ was carried out as previously described (*Brock et al., 2020*). Briefly, the locus of plasticity expression was determined by the angle $\varphi$ between the diagonal and the CV endpoint. Presynaptic expression was indicated by $\varphi>0$ at the p<0.05 level as determined using a standard one-sample *t*-test, whereas postsynaptic expression was similarly indicated by $\varphi<0$, whereas no significance would suggest a mixed pre- and

postsynaptic locus of expression (*Sjöström et al., 2007*). CV analysis was only carried out with experiments showing at least 5% plasticity, meaning EPSP after/before >105% for tLTP and EPSP after/before <95% for tLTD.

(PPR) was calculated as ($EPSP_2 - EPSP_1$)/$EPSP_1$, taken from averages before and after tLTD induction. The change in PPR was calculated as $\Delta PPR = PPR_{after} - PPR_{before}$. A change in PPR after tLTD induction suggested altered probability of release, i.e., a presynaptic locus of expression (*Sjöström et al., 2003*).

As a clarification regarding the use of different baseline spiking patterns, we note that the presence of preNMDARs complicates the use of paired-pulse stimulation during baseline periods, since preNMDARs enhance release during high-frequency activity (*Abrahamsson et al., 2017*; *Sjöström et al., 2003*; *Wong et al., 2024*). Therefore, repeated stimulation can suppress synaptic responses when preNMDARs are blocked, potentially confounding interpretation. For this reason, we limited PPR analysis to *Figures 5 and 6*, where conditions were appropriate.

## Pharmacology

MK-801 maleate (Hello Bio) and 7-CK (Alomone Labs) were washed in at 2 mM and 100 µM, respectively (*Nabavi et al., 2013*; *Sjöström et al., 2003*), and slices were incubated for at least 30 min before the start of recordings. We used this high MK-801 concentration to match the mM-range concentrations commonly used intracellularly (*Berretta and Jones, 1996*; *Brasier and Feldman, 2008*; *Buchanan et al., 2012*; *Corlew et al., 2007*; *Larsen et al., 2011*; *Nevian and Sakmann, 2006*; *Rodríguez-Moreno et al., 2011*; *Rodríguez-Moreno and Paulsen, 2008*), allowing a direct comparison of extra/intracellular MK-801 application. Lower extracellular MK-801 concentrations in the µM range (e.g. *Huettner and Bean, 1988*; *Kemp et al., 1988*; *Tovar and Westbrook, 1999*) were thus avoided to ensure robust NMDAR blockade and avoid false negatives due to incomplete inhibition. In JNK2 blockade experiments, slices were incubated in ACSF containing 4 µM SP600125 (Sigma-Aldrich) (*Bennett et al., 2001*) for at least 2 hr before the start of recordings. This concentration is specific for JNK2 over JNK1 and JNK3 (*Abrahamsson et al., 2017*; *Nisticò et al., 2015*).

The peptide used to selectively disrupt JNK2/STX1a interaction was synthesized by Université de Sherbrooke and corresponds to 12 residues (IEQSIEQEEGLNRS) that are part of the N-terminal amino acid sequence of STX1a interacting with JNK2 (*Marcelli et al., 2019*). Patch pipettes were loaded with 10 µM of the peptide. Neurons were patched for at least 30 min before tLTD induction.

## Statistics

Unless stated otherwise, results are reported as the mean ± standard error of the mean, with n indicating the number of connections. Boxplots show medians and quartiles, with whiskers denoting extremes. Significance levels are denoted using asterisks (*$p<0.05$, **$p<0.01$, ***$p<0.001$). Pairwise comparisons were carried out using a two-tailed Student's *t*-test for equal means, unless otherwise indicated. If the equality of variances *F*-test gave $p<0.05$, we employed the unequal variances *t*-test. Wilcoxon-Mann-Whitney's non-parametric test was used in parallel, with a similar outcome to the *t*-test. Comparisons to a single value were done with a one-sample *t*-test, e.g., for CV analysis $\varphi$. Multiple comparisons were carried out using one-way ANOVA with Bonferroni's post-hoc correction. Pairwise comparisons were only made if ANOVA permitted it at the $p<0.05$ level. Based on the outcome of Bartlett's test, we used homo- or heteroscedastic (Welch) ANOVA. Statistical tests were performed in Igor Pro and Prism 7.0 (GraphPad Software).

## Acknowledgements

We thank Alanna Watt, Kim Dore, Airi Watanabe, and members of the Sjöström lab for their help and useful discussions. We thank Marc-André Bonin for help with JGRi1 peptide synthesis. AT was supported by the Marie Skłodowska-Curie fellowship 892837. SR was supported by doctoral awards from FRQS (317516), HBHL, and the RI-MUHC. JAB won Max Stern and IPN recruitment awards. HH-WW was supported by CIHR fellowship 295104, an HBHL postdoctoral fellowship, FRQS postdoctoral fellowship 259572, and QBIN scholarship 35450. PJS was a recipient of CFI LOF 28331, CIHR PG 156223, 191969, 191997, FRSQ CB 254033, NSERC DG/DAS 2024–06712, 2017–04730, 2017–507818, and Donald S Wells Distinguished Scientist awards. The funders had no role in study design, data collection and interpretation, or the decision to submit the work for publication.

## Additional information

### Funding

| Funder | Grant reference number | Author |
|---|---|---|
| Marie Skłodowska-Curie Actions | 10.3030/892837 | Aurore Thomazeau |
| Fonds de Recherche du Québec - Santé | 317516 | Sabine Rannio |
| McGill University | HBHL | Sabine Rannio Hovy Ho-Wai Wong |
| RI-MUHC | | Sabine Rannio |
| McGill University | Max Stern recruitment award | Jennifer A Brock |
| McGill University | IPN recruitment award | Jennifer A Brock |
| Canadian Institutes of Health Research | 295104 | Hovy Ho-Wai Wong |
| Fonds de Recherche du Québec - Santé | 259572 | Hovy Ho-Wai Wong |
| QBIN | 35450 | Hovy Ho-Wai Wong |
| Canada Foundation for Innovation | 28331 | Per Jesper Sjöström |
| Canadian Institutes of Health Research | 156223 | Per Jesper Sjöström |
| Canadian Institutes of Health Research | 191969 | Per Jesper Sjöström |
| Canadian Institutes of Health Research | 191997 | Per Jesper Sjöström |
| Fonds de Recherche du Québec - Santé | 254033 | Per Jesper Sjöström |
| Natural Sciences and Engineering Research Council of Canada | 2024-06712 | Per Jesper Sjöström |
| Natural Sciences and Engineering Research Council of Canada | 2017-04730 | Per Jesper Sjöström |
| Natural Sciences and Engineering Research Council of Canada | 2017-507818 | Per Jesper Sjöström |
| Donald S. Wells Distinguished Scientist award | | Per Jesper Sjöström |

The funders had no role in study design, data collection and interpretation, or the decision to submit the work for publication.

### Author contributions

Aurore Thomazeau, Conceptualization, Resources, Data curation, Formal analysis, Funding acquisition, Investigation, Visualization, Methodology, Writing – original draft, Project administration, Writing – review and editing; Sabine Rannio, Conceptualization, Data curation, Formal analysis, Supervision, Funding acquisition, Validation, Investigation, Visualization, Methodology, Writing – original draft, Writing – review and editing; Jennifer A Brock, Hovy Ho-Wai Wong, Conceptualization, Data curation, Formal analysis, Funding acquisition, Investigation, Methodology; Per Jesper Sjöström, Conceptualization, Resources, Data curation, Software, Formal analysis, Supervision, Funding acquisition, Validation, Visualization, Methodology, Writing – original draft, Project administration, Writing – review and editing

## Author ORCIDs

Aurore Thomazeau (ID) https://orcid.org/0000-0002-7668-2867
Sabine Rannio (ID) https://orcid.org/0000-0003-0669-3680
Hovy Ho-Wai Wong (ID) http://orcid.org/0000-0003-3317-478X
Per Jesper Sjöström (ID) https://orcid.org/0000-0001-7085-2223

## Ethics

The animal study was reviewed and approved by the Montreal General Hospital Facility Animal Care Committee (The MGH FACC) as Animal Use Protocol (AUP) 6041 and strictly adhered to the guidelines of the Canadian Council on Animal Care (CCAC).

Reviewer #2 (Public review): https://doi.org/10.7554/eLife.106284.3.sa1
Reviewer #3 (Public review): https://doi.org/10.7554/eLife.106284.3.sa2
Author response https://doi.org/10.7554/eLife.106284.3.sa3

## Additional files

### Supplementary files

MDAR checklist

### Data availability

The complete dataset is available on Dryad: https://doi.org/10.5061/dryad.3n5tb2rwb.

The following dataset was generated:

| Author(s) | Year | Dataset title | Dataset URL | Database and Identifier |
|---|---|---|---|---|
| Aurore T, Sabine R, Jennifer BA, Hovy Ho-Wai W, Jesper SP | 2025 | Neocortical Layer-5 tLTD Relies on Non-Ionotropic Presynaptic NMDA Receptor Signaling | https://doi.org/10.5061/dryad.3n5tb2rwb | Dryad Digital Repository, 10.5061/dryad.3n5tb2rwb |

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
