## [Editor Report · eLife Assessment]

By using sparse Cre-dependent deletion of GluN1 subunit, in vitro quadruple patch clamp recordings, and pharmacological interventions, the authors show that spike timing dependent plasticity at between L5 synapses in the mouse visual cortex is: (i) dependent on presynaptic NMDA receptors; (ii) mediated by non-ionotropic NMDA receptor signaling, and (iii) reliant on presynaptic JNK2/Syntaxin-1a interactions. These **fundamental** findings advance our understanding of the molecular mechanisms underlying spike time dependent plasticity. The data are **compelling** and are supported by the elegant application of sophisticated experimental approaches.

---

## [Referee Report · Reviewer #2 (Public review)]

Summary:

The study characterized the dependence of spike timing-dependent long-term depression (tLTD) on presynaptic NMDA receptors and the intracellular cascade after NMDAR activation possibly involved in the observed decrease in glutamate probability release at L5-L5 synapses of the visual cortex in mouse brain slices.

Strengths:

The genetic and electrophysiological experiments are thorough. The experiments are well reported and mainly support the conclusions. This study confirms and extends current knowledge by elucidating additional plasticity mechanisms at cortical synapses, complementing existing literature.

Weaknesses:

No direct testing for ions passing trough standard NMDAR, mainly sodium and calcium is shown.

---

## [Referee Report · Reviewer #3 (Public review)]

Summary:

In this manuscript, "Neocortical Layer-5 tLTD Relies on Non-Ionotropic Presynaptic NMDA Receptor Signaling", Thomazeau et al. seek to determine the role of presynaptic NMDA receptors and the mechanism by which they mediate expression of frequency-independent timing-dependent long-term depression (tLTD) between layer-5 (L5) pyramidal cells (PCs) in the developing mouse visual cortex. By utilizing sophisticated methods, including sparse Cre-dependent deletion of GluN1 subunit via neonatal iCre-encoding viral injection, in vitro quadruple patch clamp recordings, and pharmacological interventions, the authors elegantly show that L5 PC->PC tLTD is (1) dependent on presynaptic NMDA receptors, (2) mediated by non-ionotropic NMDA receptor signaling, and (3) is reliant on JNK2/Syntaxin-1a (STX1a) interaction (but not RIM1αβ) in the presynaptic neuron. The study elegantly and pointedly addresses a long-standing conundrum regarding the lack of frequency dependence of tLTD.

Strengths:

The authors did a commendable job presenting a very polished piece of work with high-quality data that this Reviewer feels enthusiastic about. The manuscript has several notable strengths. Firstly, the methodological approach used in the study is highly sophisticated and technically challenging, and successfully produced high-quality data that were easily accessible to a broader audience. Secondly, the pharmacological interventions used in the study targeted specific players and their mechanistic roles, unveiling the mechanism in question step-by-step. Lastly, the manuscript is written in a well-organized manner that is easy to follow. Overall, the study provides a series of compelling evidence that leads to a clear illustration of mechanistic understanding.

Weakness:

No major weaknesses were noted.

---

## [Author Response]

The following is the authors’ response to the original reviews

**Reviewer 1 (Public review)**
SummaryThe results offer compelling evidence that L5-L5 tLTD depends on presynaptic NMDARs, a concept that has previously been somewhat controversial. It documents the novel finding that presynaptic NMDARs facilitate tLTD through their metabotropic signaling mechanism.

We thank Reviewer 1 for their kind words and thoughtful feedback!

StrengthsThe experimental design is clever and clean. The approach of comparing the results in cell pairs where NMDA is deleted either presynaptically or postsynaptically is technically insightful and yields decisive data. The MK801 experiments are also compelling.

We are very grateful for this kind feedback!

WeaknessesNo major weaknesses were noted by this reviewer.

We were happy to see that Reviewer 1 had no concerns in the Public Review. We address their Recommendations here below.

**Reviewer #1 (Recommendations for the authors):**
There is one minor issue that the authors might want to address. In Figure 6C, the average time course of the controls (blue symbols) shows a clear decline in the baseline. The rate of this decline appears to be similar to the initial decline rate observed after inducing tLTD.

Sorry, the x-axis was truncated so the first data points were not visible. We fixed Fig 6C as well as 6G, which suffered from the same problem.

**Reviewer 2 (Public review)**
SummaryThe study characterized the dependence of spike-timing-dependent long-term depression (tLTD) on presynaptic NMDA receptors and the intracellular cascade after NMDAR activation possibly involved in the observed decrease in glutamate probability release at L5-L5 synapses of the visual cortex in mouse brain slices.

We are grateful for Reviewer 2’s thoughtful and detailed feedback!

StrengthsThe genetic and electrophysiological experiments are thorough. The experiments are well-reported and mainly support the conclusions. This study confirms and extends current knowledge by elucidating additional plasticity mechanisms at cortical synapses, complementing existing literature.

We were thrilled to see that the reviewer thinks our experiments are “thorough”, “well-reported” and they “mainly support the conclusions”!

WeaknessesWhile one of the main conclusions (preNMDARs mediating presynaptic LTD) is resolved in a very convincing genetic approach, the second main conclusion of the manuscript (non-ionotropic preNMDARs) relies on the use of a high concentration of extracellular blockers (MK801, 2 mM; 7-clorokinurenic acid: 100 microM), but no controls for the specific actions of these compounds are shown.

We thank the reviewer for calling our genetic approach “very convincing”!

Regarding the pharmacological controls: for MK-801, we deliberately used a high extracellular concentration in the mM-range to match the intracellular concentrations used both in our own experiments and in prior studies (Berretta and Jones, 1996; Brasier and Feldman, 2008; Buchanan et al., 2012; Corlew et al., 2007; Humeau et al., 2003; Larsen et al., 2011; Rodríguez-Moreno et al., 2011; Rodríguez-Moreno and Paulsen, 2008). Our goal was to isolate the variable of application site (internal vs. external) while keeping concentration constant. If we had used the lower, more conventional µM-range extracellular concentrations (e.g., Huettner and Bean, 1988; Kemp et al., 1988; Tovar and Westbrook, 1999), differences in outcome might have reflected differences in drug efficacy rather than localization — particularly since failure to observe an effect at low concentrations would be hard to interpret.

We now clarify this rationale in the revised manuscript (lines 578-585).

As for 7-chlorokynurenic acid (7-CK), the 100 µM concentration we used is standard for effectively blocking the glycine-binding site of NMDARs (e.g., Nabavi et al., 2013).

We also added two supplementary figures to show the effects of washing in MK-801 and 7-CK. In MK-801, responses are stable at low frequency (clarified in the manuscript lines 155-157 and Supp Fig 1 caption text). However, 7-CK suppresses responses appreciably, which takes time to stabilize. We clarify in the revised manuscript that in 7-CK experiments, we waited for this stabilization before inducing tLTD (lines 167-172 and Supp Fig 2 caption text). This additional suppression is consistent with 7-CK also acting as a potent competitive inhibitor of L-glutamate transport into synaptic vesicles (Bartlett et al., 1998).

In addition, no direct testing for ions passing through preNMDAR has been performed.

Sorry for being unclear, we have previously tested directly for ions passing through preNMDARs. For example, we showed blockade with Mg^2+^ before (Abrahamsson et al., 2017; Wong et al., 2024), and we showed preNMDAR Ca^2+^ supralinearities before (Abrahamsson et al., 2017; Buchanan et al., 2012). To improve the manuscript, we clarified the text accordingly (lines 140-141).

It is not known if the results can be extrapolated to adult brain as the data were obtained from 11-18 days-old mice slices, a period during which synapses are still maturing and the cortex is highly plastic.

Thank you, this is a good point. We address this point in the revised manuscript (lines 428-432). While our study focuses on the early postnatal period (P11–P18), when plasticity mechanisms are prominent and synaptic maturation is ongoing, we agree that extrapolation to the adult brain should be made with caution.

**Reviewer #2 (Recommendations for the authors):**

Points 1-3 were also found in the Public Review so are not addressed again here.

(4) Results seem to be obtained in the absence of inhibition blocking and the role of inhibition in tLTD is not described. It should be indicated whether present results are obtained with or without the functional inhibitory synapse activation. If GABAergic synapses are not blocked authors need to show what happens when this inhibition is blocked.

We agree that extracellular stimulation can inadvertently recruit inhibitory circuits. However, in our paired whole-cell recordings, synaptic responses are always subthreshold and exclusively reflect the direct connection between the two recorded neurons (Chou et al., 2024; Song et al., 2005). Under these conditions, inhibitory synapses are not activated, and we therefore did not apply GABAergic blockers. We thank the reviewer for raising this, which is now clarified in the Methods (lines 539-541) of the revised manuscript.

(5) In some figures, the number of experiments seems to be low, and this number of experiments might be increased (Figures 1C, 3C, 4B).

We acknowledge that the number of experiments in these figures is modest, but these recordings are technically demanding, and the data are carefully curated. Importantly, the observed effects were statistically significant, indicating that the sample sizes were sufficient. We also note that concerns about statistical power are typically more critical in the case of negative or null results, whereas our findings were positive.

(6) The discussion is detailed but it is not clear that the activation of JNK2 needs to be achieved by a non-ionotropic action of NMDAR as activation after ionotropic NMDAR activation has been described in the literature. This point needs to be clarified and expanded.

Sorry that we were unclear on this point. We clarified this on lines 371-372 of the manuscript.

(7) Adding a cartoon/schematic summarizing the proposed mechanism for tLTD would help the reading of the manuscript.

We appreciate this suggestion and agree that a schematic would be helpful. However, we prefer to hold off on including one at this stage, as aspects of the underlying mechanism — particularly the role of CB1 receptors in presynaptic pyramidal cells (Sjöström et al., 2003) — are currently under active investigation in a separate project. To avoid potentially misleading oversimplifications, we would prefer to revisit a summary schematic once these uncertainties have been resolved.

Minor:(1) Concentration of compounds is recommended to be included in the figures or in the text. This would make it easy to follow the results.

We appreciate the suggestion. However, we avoid repeating concentrations to emphasize that conditions are consistent unless otherwise stated. All compound concentrations are clearly listed in the Methods and remain unchanged across experiments. We believe this streamlined approach avoids redundancy while keeping the results clear.

(2) In some figures, failures in synaptic transmission can be observed (and changes after tLTD). The authors may analyse changes in a number of failures in synaptic transmission after tLTD as an additional indication of a presynaptic expression of this form of tLTD. PPR may also be included in all figures.

While failures in synaptic transmission are occasionally visible, we chose to focus on CV analysis, which is mathematically equivalent to failure rate analysis, as both rely on the same underlying variability in synaptic responses (Brock et al., 2020). Provided failures are reliably extracted (which requires sufficient signal-to-noise), CV and failure rate analyses should yield consistent conclusions.

In contrast, PPR analysis is not mathematically equivalent to CV analysis and may offer complementary insights into presynaptic mechanisms. However, the presence of preNMDARs complicates the use of paired-pulse stimulation during baseline: preNMDARs enhance release during high-frequency activity (Abrahamsson et al., 2017; Sjöström et al., 2003; Wong et al., 2024), so repeated stimulation can suppress synaptic responses when preNMDARs are blocked, potentially confounding interpretation. For this reason, we limited PPR analysis to Figures 5 and 6, where conditions were appropriate.

Admittedly, our manuscript was previously not clear on when we did paired-pulse stimulation and when we did not. We have clarified this in the revised manuscript (lines 548- 551 and lines 569-574).

(3) Discussion: Line 363-64, hippocampal (SC-CA1 synapses) results exist where postsynaptic MK801 blocks presynaptic tLTD, this may be added here and in the references.

While we acknowledge that postsynaptic MK-801 has been shown to block presynaptic tLTD at hippocampal SC–CA1 synapses, we note that the hippocampus is part of the archicortex, whereas our study focuses on neocortical circuits, as highlighted in the manuscript title. Given the substantial anatomical and functional differences between these regions, we prefer to keep our discussion focused on the neocortex to maintain conceptual coherence.

(4) Discussion: While authors indicate "non-ionotropic" they do not discuss whether this action can be named properly "metabotropic" and whether G-proteins may be in fact needed for this action. The authors may briefly discuss this point.

We previously referred to non-ionotropic NMDAR signaling as “metabotropic,” but reconsidered after discussions with colleagues, including Juan Lerma, who pointed out that the term typically implies G-protein coupling, which has not been definitively shown in this context. While the term “metabotropic” is used inconsistently in the literature (Heuss and Gerber, 2000; Heuss et al., 1999) — sometimes broadly to indicate non-ion flow signaling — we prefer to avoid potential confusion and therefore use “non-ionotropic” unless and until G-protein involvement is clearly demonstrated. We clarified this on lines 423-427 of the Discussion.

(5) Page 19, line 451 NMDR needs to be corrected to NMDAR.

Thanks! This was corrected.

**Reviewer 3 (Public review)**
SummaryIn this manuscript, "Neocortical Layer-5 tLTD Relies on Non-Ionotropic Presynaptic NMDA Receptor Signaling", Thomazeau et al. seek to determine the role of presynaptic NMDA receptors and the mechanism by which they mediate expression of frequency-independent timing-dependent long-term depression (tLTD) between layer-5 (L5) pyramidal cells (PCs) in the developing mouse visual cortex. By utilizing sophisticated methods, including sparse Cre-dependent deletion of GluN1 subunit via neonatal iCre-encoding viral injection, in vitro quadruple patch clamp recordings, and pharmacological interventions, the authors elegantly show that L5 PC->PC tLTD is (1) dependent on presynaptic NMDA receptors, (2) mediated by non-ionotropic NMDA receptor signaling, and (3) is reliant on JNK2/Syntaxin-1a (STX1a) interaction (but not RIM1αβ) in the presynaptic neuron. The study elegantly and pointedly addresses a long-standing conundrum regarding the lack of frequency dependence of tLTD.

We thank the reviewer for calling our methods “sophisticated” and our study “elegant”! We appreciate the kind feedback!

StrengthsThe authors did a commendable job presenting a very polished piece of work with high-quality data that this Reviewer feels enthusiastic about. The manuscript has several notable strengths. Firstly, the methodological approach used in the study is highly sophisticated and technically challenging and successfully produced high-quality data that were easily accessible to a broader audience. Secondly, the pharmacological interventions used in the study targeted specific players and their mechanistic roles, unveiling the mechanism in question step-by-step. Lastly, the manuscript is written in a well-organized manner that is easy to follow. Overall, the study provides a series of compelling evidence that leads to a clear illustration of mechanistic understanding.

We are elated that the reviewer described our study with words such as “polished”, “high-quality”, “sophisticated”, and “compelling”!

Minor comments(1) For the broad readership, a brief description of JNK2-mediated signaling cascade underlying tLTD, including its intersection with CB1 receptor signaling may be desired.

Thank you, this is a great suggestion for improving clarity. We briefly address this point in the revised manuscript (lines 360-363).

(2) The authors used juvenile mice, P11 to P18 of age. It is a typical age range used for plasticity experiments, but it is also true that this age range spans before and after eye-opening in mice (~P13) and is a few days before the onset of the classical critical period for ocular dominance plasticity in the visual cortex. Given the mechanistic novelty reported in the study, can authors comment on whether this signaling pathway may be age-dependent?

Thanks, Reviewer 2 also raised this point. In the revised manuscript, we discuss this point (lines 428-432).

**Reviewer #3 (Recommendations for the authors):**
(1) Minor typos: page 4 line 101: sensitivity -> sensitive.

We fixed this typo.

(2) Page 15 line 333: sensitivity -> sensitive.

We fixed this typo.

(3) Minor aesthetic suggestion: On the scale bars for all examples, LTP and LTD data are easily confused with the letter L. I'd suggest flipping them left to right.

We thank the reviewer for the suggestion. We flipped the scale bars in all figures.

References

Abrahamsson, T., Chou, C.Y.C., Li, S.Y., Mancino, A., Costa, R.P., Brock, J.A., Nuro, E., Buchanan, K.A., Elgar, D., Blackman, A.V., et al. 2017. Differential Regulation of Evoked and Spontaneous Release by Presynaptic NMDA Receptors. Neuron 96: 839-855 e835

Bartlett, R.D., Esslinger, C.S., Thompson, C.M., and Bridges, R.J. 1998. Substituted quinolines as inhibitors of L-glutamate transport into synaptic vesicles. Neuropharmacology 37: 839-846

Berretta, N., and Jones, R.S. 1996. Tonic facilitation of glutamate release by presynaptic N-methyl-D-aspartate autoreceptors in the entorhinal cortex. Neuroscience 75: 339-344.

Brasier, D.J., and Feldman, D.E. 2008. Synapse-specific expression of functional presynaptic NMDA receptors in rat somatosensory cortex. J Neurosci 28: 2199-2211

Brock, J.A., Thomazeau, A., Watanabe, A., Li, S.S.Y., and Sjöström, P.J. 2020. A Practical Guide to Using CV Analysis for Determining the Locus of Synaptic Plasticity. Frontiers in Synaptic Neuroscience 12:11 10.3389/fnsyn.2020.00011

Buchanan, K.A., Blackman, A.V., Moreau, A.W., Elgar, D., Costa, R.P., Lalanne, T., Tudor Jones, A.A., Oyrer, J., and Sjöström, P.J. 2012. Target-Specific Expression of Presynaptic NMDA Receptors in Neocortical Microcircuits. Neuron 75: 451-466

Chou, C.Y.C., Wong, H.H.W., Guo, C., Boukoulou, K.E., Huang, C., Jannat, J., Klimenko, T., Li, V.Y., Liang, T.A., Wu, V.C., and Sjöström, P.J. 2024. Principles of visual cortex excitatory microcircuit organization. The Innovation 6: 1-11

Corlew, R., Wang, Y., Ghermazien, H., Erisir, A., and Philpot, B.D. 2007. Developmental switch in the contribution of presynaptic and postsynaptic NMDA receptors to long-term depression. J Neurosci 27: 9835-9845

Heuss, C., and Gerber, U. 2000. G-protein-independent signaling by G-protein-coupled receptors. Trends in Neurosciences 23: 469-475

Heuss, C., Scanziani, M., Gähwiler, B.H., and Gerber, U. 1999. G-protein-independent signaling mediated by metabotropic glutamate receptors. Nature Neuroscience 2: 1070-1077

Huettner, J.E., and Bean, B.P. 1988. Block of N-methyl-D-aspartate-activated current by the anticonvulsant MK-801: selective binding to open channels. PNAS 85: 1307-1311.

Humeau, Y., Shaban, H., Bissière, S., and Lüthi, A. 2003. Presynaptic induction of heterosynaptic associative plasticity in the mammalian brain. Nature 426: 841-845

Kemp, J.A., Foster, A.C., Leeson, P.D., Priestley, T., Tridgett, R., Iversen, L.L., and Woodruff, G.N. 1988. 7-Chlorokynurenic acid is a selective antagonist at the glycine modulatory site of the N-methyl-D-aspartate receptor complex. PNAS 85: 6547-6550

Larsen, R.S., Corlew, R.J., Henson, M.A., Roberts, A.C., Mishina, M., Watanabe, M., Lipton, S.A., Nakanishi, N., Perez-Otano, I., Weinberg, R.J., and Philpot, B.D. 2011. NR3A-containing NMDARs promote neurotransmitter release and spike timing-dependent plasticity. Nat Neurosci 14: 338-344

Nabavi, S., Kessels, H.W., Alfonso, S., Aow, J., Fox, R., and Malinow, R. 2013. Metabotropic NMDA receptor function is required for NMDA receptor-dependent long-term depression. PNAS 110: 4027-4032

Rodríguez-Moreno, A., Kohl, M.M., Reeve, J.E., Eaton, T.R., Collins, H.A., Anderson, H.L., and Paulsen, O. 2011. Presynaptic induction and expression of timing-dependent long-term depression demonstrated by compartment-specific photorelease of a use-dependent NMDA receptor antagonist. J Neurosci 31: 8564-8569

Rodríguez-Moreno, A., and Paulsen, O. 2008. Spike timing-dependent long-term depression requires presynaptic NMDA receptors. Nat Neurosci 11: 744-745

Sjöström, P.J., Turrigiano, G.G., and Nelson, S.B. 2003. Neocortical LTD via coincident activation of presynaptic NMDA and cannabinoid receptors. Neuron 39: 641-654

Song, S., Sjöström, P.J., Reigl, M., Nelson, S., and Chklovskii, D.B. 2005. Highly nonrandom features of synaptic connectivity in local cortical circuits. PLoS biology 3: e68

Tovar, K.R., and Westbrook, G.L. 1999. The incorporation of NMDA receptors with a distinct subunit composition at nascent hippocampal synapses in vitro. J Neurosci 19: 4180-4188

Wong, H.H., Watt, A.J., and Sjöström, P.J. 2024. Synapse-specific burst coding sustained by local axonal translation. Neuron 112: 264-276 e266